# Quantifiers in Natural Language: Efficient Communication and Degrees of Semantic Universals

**DOI:** 10.3390/e23101335

**Published:** 2021-10-14

**Authors:** Shane Steinert-Threlkeld

**Affiliations:** Department of Linguistics, University of Washington, Seattle, WA 98195, USA; shanest@uw.edu

**Keywords:** semantic universals, efficient communication, quantifiers, monotonicity, conservativity, simplicity, informativeness, semantic typology

## Abstract

While the languages of the world vary greatly, they exhibit systematic patterns, as well. Semantic universals are restrictions on the variation in meaning exhibit cross-linguistically (e.g., that, in all languages, expressions of a certain type can only denote meanings with a certain special property). This paper pursues an efficient communication analysis to explain the presence of semantic universals in a domain of function words: quantifiers. Two experiments measure how well languages do in optimally trading off between competing pressures of simplicity and informativeness. First, we show that artificial languages which more closely resemble natural languages are more optimal. Then, we introduce information-theoretic measures of degrees of semantic universals and show that these are not correlated with optimality in a random sample of artificial languages. These results suggest both that efficient communication shapes semantic typology in both content and function word domains, as well as that semantic universals may not stand in need of independent explanation.

## 1. Introduction

While the languages of the world vary greatly, linguists have discovered many restrictions on possible variation [1,2,3], at all levels of linguistic analysis. For example, when it comes to phonology, it has been argued that all languages have at least one unrounded and one back vowel [2]. Similarly, at the level of syntax, certain categories (e.g., nouns and verbs) have been argued to exist in all languages [1]. (See Reference [4] for some hesitance about positing universal syntactic categories.) It has also been argued that dependency lengths in natural language are much shorter than chance [5,6]. There are also universals of word order, which have recently been argued to arise from communicative efficiency pressures, of the kind we will discuss below [7]. (The tradition of identifying and explaining *linguistic laws* (e.g., generalizations of Zipf’s law [8]) also could be considered under the umbrella of statistical universals [9,10,11,12,13,14]. These will be discussed in more detail in Section 6.2).

Semantic universals are restrictions on the range of variation in meaning across languages. For example, while the basic color terms of the languages of the world vary greatly [15], it has been shown that all such terms denote *convex* regions of color space [16,17]. Recently, in several domains—e.g., kinship terms, color terms—such universals have been argued to arise from pressures for efficient communication, namely a trade-off between simplicity and informativeness [18,19,20]. Roughly: a language cannot be both maximally simple (in terms of, e.g., cognitive load) and at the same time maximally informative (in terms of, e.g., helping a speaker convey their intended meaning to a listener). Intuitively, a maximally simple language would have a single term, which could not be used to convey significant information. A maximally informative language, on the other hand, would contain individual expressions for every possible thought to be expressed; such a language would be highly complex, relying on significant memorization. (These concepts can and will be made precise in information-theoretic terms [20,21]. For instance, the maximally informative language described here maximizes the mutual information between words and meanings. Our precise definitions are in Section 3). The general claim: the semantic systems of the world’s languages optimally balance these two competing pressures. (See Reference [9] for the closely related idea that word frequency distributions arise from competing pressures of minimizing effort for the speaker and the hearer.)

While the aforementioned case studies apply to domains of content words, the historically most prominent domain of semantic universals has been from a domain of function words, namely determiners [22,23]. In particular, the quantifiers expressed by determiners have been argued to have properties, such as monotonicity, quantitativeness, and conservativity. Recent work has offered a different explanation for these universals in quantifiers, namely that they arise from a pressure of learnability: quantifiers satisfying the universals are easier to learn than those that do not and, therefore, get lexicalized [24,25]. This argument, however, does not rule out the possibility of explaining these universals in terms of the aforementioned trade-off.

In this paper, we present two experiments which suggest that the semantic universals for quantifiers can be seen as arising from the trade-off between simplicity and informativeness. Both experiments measure how optimally a language trades off these two pressures, and whether ‘more natural’ languages are more optimal. The first experiment introduces a measure of the *degree of naturalness* of an artificial language as the percentage of quantifiers in the language which belong to those known to exist in natural languages. This measure significantly correlates with optimality. (This is a replication of the main result reported in Reference [26]). For the second experiment, we introduce information-theoretic measures of *degrees of semantic universals*—namely, monotonicity and conservativity—and show that, surprisingly, these measures are not correlated with optimality in a random sample of languages. That is: artificial languages that have a higher degree of these universals do not tend to balance these pressures any better than those with lower degrees.

These results suggest at least two conclusions. On the one hand, the first experiment shows that the semantic typology of quantifiers, a functional part of the lexicon, can be explained by efficient communication. Building on Reference [26], this is the first of a growing number of results suggesting that the same pressures shape semantic systems in domains of function words in addition to content words, from indefinites [27] to logical connectives [28,29] to person systems [30]. More provocatively, the two experimental results, taken together, suggest that semantic universals may be epiphenomenal: while empirically true, they arise as a consequence of a fundamental pressure for efficient communication and do not themselves stand in need of independent explanation.

The paper is structured as follows. In the next section, we introduce relevant background in quantifiers, their semantic universals, and the efficient communication hypothesis. Section 3 introduces methods common to both experiments: how to measure simplicity and informativeness for quantifiers, as well as the optimality of a language. Section 4 presents the results of the first experiment, while Section 5 presents the second experiment, with degrees of the semantic universals defined in Section 5.1. Following that, a discussion section explores the consequences of these two experimental results and points to directions for future work.

## 2. Background

### 2.1. Quantifier Semantics

The semantic domain we focus on is that of quantifiers, which are the semantic objects expressed by determiners. A determiner is an expression taking a common noun as an argument and generating a Noun Phrase [22,23]. We will assume, following Reference [22], a division of the determiners into two classes: grammatically simple and complex. Examples of simple determiners are *all*, *some*, *no*, *few*, *most*, *five*. Examples of complex determiners are *all but five*, *fewer than three*, *at least eight*, or *fewer than five*. Note that we do not at present provide a full account of exactly what the distinction amounts to. For example, while being a single ‘word’ (i.e., monomorphemic) certainly suffices for being simple, we leave it open that some determiners that are not monomorphemic will still count as simple. (Arguably, *most* is not monomorphemic. See References [31,32,33,34]. Moreover, some argue that a much wider class, including *no* and *few*, are also not monomorphemic. However, these arguably should count as simple for the purpose of formulating semantic universals.) For present purposes, however, the reader may consider the simple determiners to be the monomorphemic ones (roughly: single words).

As a first approximation, and following the influential study of Reference [22], we assume that determiners denote what are called type 〈1,1〉 generalized quantifiers. In other words: quantifiers are the mathematical object denoted by the determiners, which are syntactic units in a natural language. These quantifiers are relations between two subsets of a given domain of discourse. For example, using the notation where 〚·〛 is the semantic interpretation function that maps expressions of a natural language to their meaning:$$\begin{array}{ccc} 〚\mathrm{every}〛 & = & \left\{\langle M,A,B\rangle :\;A\subseteq B\right\}\\ 〚\mathrm{at}\_\mathrm{most}\_3〛 & = & \left\{\langle M,A,B\rangle :\;|A\cap B|\leq 3\right\}\\ 〚\mathrm{most}〛 & = & \left\{\langle M,A,B\rangle :\;|A\cap B|>|A\backslash B|\right\}. \end{array}$$

Moving forward, we will refer to a given tuple M:=〈M,A,B〉 as a model (and will use the term ‘structure’ interchangeably), so that a quantifier is a set of models. To see an example, the sentence ‘Every person is happy’ will be true just in case the model M:=〈M,〚person〛,〚happy〛〉∈〚every〛, which according to the definition, is just when 〚person〛⊆〚happy〛, i.e., the set of people is a subset of the set of happy entities. (Here, *M* is the ‘domain of discourse’, the set of objects relevant for a given conversation.)

As a shorthand, we will say that a determiner has a certain semantic property to mean that the quantifier that the determiner denotes has that property. Sometimes, for a determiner, such as *every*, we will write every as a shorthand for 〚every〛, i.e., for the *quantifier* that it denotes. We will use Q and its ilk as variables over quantifiers. Because quantifiers are viewed as set-theoretic objects, we will write M∈Q when a structure/model M belongs to a quantifier. (See Reference [23] for a thorough exposition of quantifiers in this tradition.) In other words, when a sentence *Det N VP* is true when interpreted in a model M, we will write 〈M,〚N〛,〚VP〛〉∈Det. In the experiments that follow below, we will take a *language* to be a set of quantifiers.

### 2.2. Semantic Universals for Quantifiers

Having now introduced the framework of generalized quantifiers for the semantics of natural language determiners, in this section, we will introduce and precisely define the two semantic universals studies in this paper: monotonicity and conservativity [22,23,35]. Both of these properties state that the (grammatically) *simple* determiners in natural languages only ever denote a mathematically distinguished subset of the possible quantifiers. One can think of this in terms of a dictionary: the simple determiners are single words, which will have entries in a dictionary. The universals state that only particular kinds of quantifiers get entered as entries in the dictionary (i.e., lexicalized); languages will rely on more complex expressions and compositional interpretation to express other quantifiers [23,24].

#### 2.2.1. Monotonicity

To motivate our first universal, consider the following sentences.
(1)a. Many scientists program in Python.b. Many scientists program.


It is clear that (1a) entails (1b): the former cannot be true without the latter being true. Similarly, this entailment does not depend on the choice of the restrictor—*scientists*—or nuclear scopes—*program in Python* and *program*—so long as the latter scope is strictly more general than the former. Moreover, competent speakers of English recognize this fact easily. What speakers thereby implicitly know is that *many* is upward monotone:
(2)Q is *upward monotone* if and only if whenever 〈M,A,B〉∈Q and $B\subseteq B^{\prime }$; then, $\langle M,A,B^{\prime }\rangle \in Q$.


By contrast, the pattern reverses if we replace *many* with *few*, as seen in the following examples.
(3)a. Few scientists program in Python.b. Few scientists program.


Here, (3b) entails (3a). Now, truth is preserved when we move from a more general scope to a more specific scope. In this case, we say that *few* is downward monotone:
(4)Q is *downward monotone* if and only if whenever 〈M,A,B〉∈Q and $B⊇B^{\prime }$; then, $\langle M,A,B^{\prime }\rangle \in Q$.


Finally, a determiner is *monotone* if and only if it is either upward or downward monotone. (Similar definitions can be given for the ‘left’ argument, i.e., the restrictor. See Section 5.1.1). The reader can verify that all of the simple determiners mentioned at the beginning of the section are monotone.

This appears to be no accident of our choice of English or of that particular list of simple determiners. Reference [22] proposed the following semantic universal.

Monotonicity Universal: All simple determiners are monotone. (Reference [22] also included conjunction of monotone quantifiers in this definition. This was, however, mainly to capture exact readings of bare numerals (e.g., *three*). Because, however, many theorists take bare numerals to have an ‘at least’ meaning, that clause is not needed.)

This universal rules out quantifiers, such as *an even number of* and *at least 6* or *at most 2*: increasing or decreasing the set *B* can cause the cardinality of A∩B to change in a way that flips the truth value of sentences with those determiners, so they are not monotone. The claim then is that no simple determiner in any natural language denotes those quantifiers.

#### 2.2.2. Conservativity

Our next universal captures the intuition that the restrictor genuinely restricts what a sentence talks about. That is, sentences of the form *Det N VP* are in some sense about the *N*s and nothing else. This universal can be observed by noting the felt equivalence between the following pairs of sentences.
(5)a. Every student passed.b. Every student is a student who passed.(6)a. Most Amsterdammers ride a bicycle to work.b. Most Amsterdammers are Amsterdammers who ride a bicycle to work.


The formal concept at play here has been called conservativity.
(7)Q is *conservative* if and only if 〈M,A,B〉∈Q if and only if 〈M,A,A∩B〉∈Q.


Reference [22] formulated and defended the following universal. (Because the term *conservative* was not introduced until Reference [35], the original formulation was in terms of a quantifier *living on* a witness set. We follow the norm of formulating in terms of conservativity for concision.)

Conservativity Universal: All simple determiners are conservative. (In fact, conservativity often is a claim about all determiners, not just the simple ones. See Reference [24] and references therein for discussion.)

This universal rules out quantifiers that depend on other portions of the model besides *A*, such as B\A. As an example, there is no determiner *equi* in any language such that the following two sentences are equivalent in meaning.
(8)a. Equi students are at the park.b. The number of students is the same as the number of people at the park.


### 2.3. The Learnability hypothesis

In recent work, semantic universals for quantifiers [24], color terms [25], and responsive predicates [36] have been argued to arise from a *learnability* pressure. Expressions satisfying these semantic universals are shown to be easier to learn than those that do not satisfy the universals. Together with the assumption that languages choose to lexicalize easier-to-learn meanings, these facts about learnability would explain the typological facts in these domains.

While the aforementioned studies all used methods from machine learning to measure learnability at a large scale, several studies have also studied the learnability of these properties in children and non-human primates. Reference [37] argues that conservative quantifiers are easier than non-conservative ones for children to learn (though, see Reference [38] for a failed replication). (Reference [24] also found no learnability difference between conservative and non-conservative quantifiers, thus discussing the possibility that conservativity has a different source.) Similarly, Reference [39] finds that monotone, as well as connected (roughly: conjunction of monotone), quantifiers are easier to learn than non-monotone ones, and Reference [40] finds that this learnability bias is present in non-human primates, as well. In addition to these, artificial language learning experiments have suggested that cross-linguistically attested systems for person marking [41] and evidentials [42] are easier to learn.

Taken together, these results provide strong evidence that learnability plays a role in shaping the semantic systems of the world’s languages.

### 2.4. The Efficient Communication Hypothesis

While the previously discussed studies show that learnability likely plays a role in shaping semantic systems, they by no means imply that it is the *only* (or even the best) explanation. An alternative account with wide appeal holds *efficient communication* to be a key explanatory principle in semantic typology. See Reference [19] for an overview and Reference [43] for a survey going beyond semantics.

The key idea here is that natural languages optimally trade-off between two competing pressures: to be simple (e.g., to represent cognitively) and to be useful in communication (i.e., informative). Intuitively, one can think of simplicity in terms of how hard it would be to store all of the meanings in a language and one can think of informativity in terms of how well a speaker can convey an intended meaning to a listener. A maximally simple language may have a single word; this would be very unhelpful for communication, since it does not allow speakers to make any distinctions. By contrast, a maximally informative language may have a unique expression for every possible meaning; this would be, however, very complex to represent or learn. (The notions of simplicity and informativeness are being used in an intuitive sense at the present moment, but they can be made precise in terms of least effort [8,9,43] or information theory [18,20,21]. Our precise definitions come in Section 3. In addition, see the discussion of linguistic laws in Section 6.2). While it is impossible to be both maximally simple and maximally informative, there will exist a *Pareto frontier* of languages which optimally trade-off the two pressures: these are the languages for which there is no other language that is both simpler and more informative. In other words, the languages on the Pareto frontier are those that are doing “as well as possible” at jointly optimizing the two competing pressures for simplicity and informativeness.

The efficient communication hypothesis states that natural languages should lie on or near this Pareto frontier. Reference [18] showed that the kinship systems of the world’s languages lie much closer to the Pareto frontier than those of artificial languages. Since that pioneering work, similar analyses have been carried out for color terms [20,44], container terms [45], and numeral systems [46]. Starting with the precursor to the present paper, Reference [26], the framework has also been applied to several domains of function words, from indefinites [27] to logical connectives [28,29] to person systems [30]. These studies suggest that efficient communication shapes the structure of the semantic systems of the world’s languages, across both content and function words.

Like Reference [27], the present paper will compare not just natural versus artificial languages but will also compare random languages that do and do not satisfy proposed semantic universals for the domain in question (indefinites there, quantifiers here). Furthermore, instead of simply comparing languages that do and do not have a property, we will introduce information-theoretic methods for measuring *degrees* of having a semantic universal, to see whether this correlates with closeness to the Pareto frontier.

## 3. Methods

While the two experiments in this paper differ in what property of languages they measure, they share many common features: the measures of simplicity and informativeness, and the measure of optimality as closeness to the Pareto frontier. In this section, we provide explicit definitions of all of these components, before turning to the actual experimental results.

### 3.1. Measuring Simplicity and Informativeness

Our measure of cognitive simplicity relies on representing quantifiers in a Language of Thought [47,48,49], i.e., using formulas in a logical language containing operations for set union, intersection, and complementation, as well as for measuring cardinalities and comparing, multiplying, and dividing them. Table 1 shows the entire set of operators used in this paper.

The *complexity* of a quantifier is the length of the *shortest* formula in this language that denotes the quantifier. We found the shortest such formula by exhaustively enumerating all formulas with up to 12 operations and comparing the truth-values across all models up to size 10. For memory reasons, we collapse isomorphic models, representing a model 〈M,A,B〉 by the cardinalities of the four sets A∩B,A\B,B\A,M\(A∪B). This prevents us from capturing quantifiers, such as *the first three*, which do not satisfy the universal known as Quantity [24]. Future work will explore methods that relax this assumption while simultaneously addressing the resulting combinatorial explosion. Even in this restricted setting, this exhaustive search generated 279,120 distinct quantifiers.

Note that using length only is equivalent to using the probability of generating an expression with a PCFG that assigns equal weight to all productions from the same non-terminal. (Future work may explore non-uniform weights in a PCFG for this domain. Ideally, these weights would be estimated from behavioral data of, e.g., human learning [49].) The complexity of a language is the sum of the complexities of the quantifiers in it. We specify an upper bound on the number of possible quantifiers in a language (10 in our experiments) and divide the sum by this number.

Our measure of informativeness stems from notions of communicative success: a speaker has an intended model that they want to communicate to a listener using the quantifiers in their language [19,50]. This is captured by the following:
$$I\left(L\right):=\sum_{M}P\left(M\right)\sum_{Q\in L}P\left(Q|M\right)\sum_{M^{\prime }\in Q}P\left(M^{\prime }|Q\right)\cdot u\left(M^{\prime },M\right).$$

The prior over models, as well as the conditional distributions, are assumed to be uniform where defined (e.g., P(Q|M)=1/n if M∈Q; 0 otherwise, where n=|{Q∈L:M∈Q}| is the number of quantifiers in *L* containing M).

This measure captures the following communicative scenario: a speaker has a model (M) in mind, that it wishes to communicate to a listener. To do so, they can use the quantifiers in the language *L*. The speaker’s behavior is captured by P(Q|M). The listener then guesses a model ($M^{\prime }$) that the speaker has in mind, with probability $P(M^{\prime }|Q)$.

The utility $u(M^{\prime },M)$ measures how good it is for the listener to guess $M^{\prime }$ when the speaker had in mind M. We base this on a measure of the distance between models, capturing the notion that non-exact matches can still be better or worse [51,52]. More precisely:
$$\begin{array}{cc} u(M^{\prime },M) & =\frac{1}{1+d(M^{\prime },M)}\;\;\;\mathrm{where}\;\\ d(M^{\prime },M) & =\sum_{X\in A\backslash B,A\cap B,B\backslash A,M\backslash (A\cup B)}\max\{0,|X|-|X^{\prime }\left|\right\}. \end{array}$$

Intuitively, this measure is inversely proportional to how many elements one has to move to transform the listener’s guessed model into the sender’s model (by summing this value across the four ‘zones’ in a model of the form 〈M,A,B〉). (The addition of 1 in the denominator both prevents division by zero and makes distance-0 models have maximal utility of 1.) For example, suppose M has 3, 4, 2, and 1 elements in A\B,A∩B,B\A,M\(A∪B), respectively, and M′ has 2, 4, 3, and 1 elements in the same zones. We then have that d(M,M′)=1, since moving one element from A\B to B\A will make the four zones have the same size in the two models.

We note here that this choice of utility function $u(M^{\prime },M)$ is one choice amongst many for defining how good of a guess $M^{\prime }$ is when M is the intended model in the speaker’s mind. Other choices are possible. For instance, one could define this utility in terms of (negative) expected surprisal of the listener in learning which sets different objects belong to. (Thanks to an anonymous reviewer for suggesting this measure.) One difficulty with implementing this is in defining distributions over the individual objects, above and beyond the distributions over models.

Furthermore, one could change the overall shape of the informativeness measure. Following Reference [18], a natural candidate would be
$$I^{\prime }\left(L\right):=\sum_{M}P\left(M\right)\sum_{Q\in L}P\left(Q|M\right)\cdot -\log P\left(M|Q\right).$$

Noting that P(M)P(Q|M)=P(M,Q), and letting M and Q be random variables jointly distributed according to *P*, one can observe that $I^{\prime }$ as just defined equals the conditional entropy H(M|Q) [21]. In information-theoretic terms, this measures how many bits of information the speaker would need to send, over and above the quantifier they chose, in order to single out their intended model. Similarly, following References [20,30], one could measure informativeness by the mutual information
$$I\left(M;Q\right):=\sum_{M,Q}P\left(M,Q\right)\log\frac{P(M|Q)}{P(M)}=H\left(M\right)-H\left(M|Q\right),$$
as well. At present, we flag this bit of modeling degree of freedom and leave the pursuit of other measures of utility (and informativeness more generally) to future work.

### 3.2. Measuring Optimality

To test whether ‘more natural’ (to be measured in different ways in the subsequent experiments) languages are more optimal, we need a measure of optimality for a language. To do this, we measure how close a language is to the *Pareto frontier*, the set of languages which are not dominated (i.e., which have no language both simpler and more informative). The Pareto frontier contains the fully optimal languages: they cannot be made less complex or more informative without becoming worse on the other dimension. Writing *P* for the Pareto frontier, we define the optimality of a language as
$$\mathrm{optimality}\left(L\right):=1-\frac{\min_{L^{\prime }\in P}d\left(L,L^{\prime }\right)}{\sqrt{2}},$$
where *d* is the Euclidean distance between points in the plane. This measure takes the *closest* point on the Pareto frontier to a given language. If a language is on the frontier, i.e., is optimal, that minimum distance will be 0, so the degree of optimality will be 1. Because both communicative cost and complexity range from 0 to 1, the theoretically largest value for the minimum distance is $\sqrt{2}$; by dividing the minimum distance by this value, optimality ranges from 0 to 1. To summarize: the degree of optimality of a language increases as it gets closer to the Pareto frontier, the set of optimal languages.

A complication arises when trying to apply this measure: because the space of possible languages is enormous, we cannot exhaustively enumerate it and thereby uncover the *true* Pareto frontier. (As noted in the previous section, there are 279,120 unique quantifiers. This means that there are ∑k=1N279120k distinct languages of *N* or fewer quantifiers (where nk is the binomial coefficient). For example, there are approximately $8\times 10^{47}$ languages of size 10 or less.) Moreover, most random sampling procedures are not guaranteed to uncover the Pareto frontier. Because of this, we need a method to estimate the Pareto frontier without being able to calculate it directly.

To estimate the true Pareto frontier, we used an evolutionary algorithm [53]. Such algorithms take inspiration from evolutionary processes in that points in a space change over a sequence of generations, with ‘children’ arising via ‘mutation’ from previous points. More importantly, such algorithms are explicitly designed to solve *multi-objective optimization* problems. Since the Pareto frontier can be seen as the set of solutions to the problem of simultaneously optimizing multiple objectives (simplicity and informativeness), these algorithms are well-suited to estimating it.

Our algorithm—provided in full detail in Algorithm A1 in Appendix A—can be intuitively described as follows. We start with an initial seed of randomly generated languages. For some specified number of ‘generations’, we select the *dominant* languages among the current set of languages. Each language then has an equal number of ‘children’ languages (enough to maintain the size of the pool of languages). A child arises from a parent language by some small sequence of ‘mutations’. In our case, this was between 1 and 3 mutations, where a mutation could be: (i) deleting a quantifier from the parent language, (ii) adding a quantifier to the parent language, or (iii) swapping a quantifier in the parent language (i.e., deleting one and adding a new one).

After running the above algorithm for some specified number of generations, we then take the dominant languages from the pool together with the languages we previously sampled, and then linearly interpolate between all of the points to form a smooth and dense frontier. (More sophisticated evolutionary algorithms specify a convergence criterion. We leave the explorations of these refinements to future work. This entire process is depicted in Figure 1.

## 4. Experiment 1: Degree of Naturalness

### 4.1. Sampling Languages

To answer the question of whether natural languages optimize the trade-off between simplicity and informativeness, we systematically control ‘how natural’ a language is by biased sampling. While a completely random language can be generated by randomly sampling a specified number of quantifiers from the space of all quantifiers generated by our grammar, an alternative sampling procedure allows us to control a degree of naturalness of sampled languages.

While there is no existing dataset of quantifiers across a large set of natural languages, a major cross-linguistic study [54,55] found that all natural language quantifiers belonged to three classes:
Generalized existential: depending only on |A∩B|.For example: 〚atleastthree〛=〈M,A,B〉:|A∩B|≥3.Generalized intersective: depending only on |A\B|.For example: 〚every〛=〈M,A,B〉:|A\B|=0.Proportional: comparing |A∩B| and |A\B|.For example: 〚most〛=〈M,A,B〉:|A∩B|>|A\B|.


We call a quantifier *quasi-natural* if it can be expressed in one of the three forms above. In addition, a language will be considered natural if it contains only quasi-natural quantifiers.

Our complete sampling procedure, then, worked as follows: for each number of words between 1 and 10, we generated 8000 languages. Each language of size *n* was chosen to have m≤n quasi-natural quantifiers, with *m* chosen uniformly from $\left\{0,\cdots ,n\right\}$. All remaining quantifiers were chosen randomly from the set of all quantifiers whose minimal formula has 12 or fewer operators. We refer to m/n as the *degree of naturalness* of a language. Thus, a language that has only quasi-natural quantifiers will have a degree of naturalness of 1 (and a language that has no quasi-natural quantifiers will have a degree of naturalness of 0).

### 4.2. Results

The main results can be seen in Figure 2. In this figure, the *x*-axis is communicative cost, which is 1−I(L), and the *y*-axis is complexity, where both I(L) and complexity are as defined in Section 3.1. Each point represents a possible language, with the color of a point corresponding to degree of naturalness. The black line is the estimated Pareto frontier, i.e., the set of languages that optimally trade-off between these two factors.

A few things can be observed right away. All of the sampled points that were found to lie on the estimated Pareto frontier (i.e., which dominate all languages both sampled and discovered by the evolutionary algorithm) appear to have a very high degree of naturalness. These are the yellow points on the black frontier, where no brown or blue points (less natural languages) are to be found. Moreover, this seems to be a general trend: it appears that languages with a high degree of naturalness tend to be closer to the Pareto frontier than those with low degrees of naturalness.

In virtue of the methods described in the previous section, we can test this appearance statistically: the Pearson correlation between optimality and degree of naturalness is ρ=0.2516, with bootstrapped 95% confidence interval [0.2444,0.2589]. Here, and in what follows, we used a standard non-parametric bootstrap, taking 10,000 bootstrap samples and estimating the confidence interval using the 2.5 and 97.5th percentiles of the empirical distributions of the samples [56]. (Appendix B shows a plot of optimality versus naturalness directly.) This significant positive correlation can be interpreted as follows: as languages become more similar to natural languages with respect to their quantifiers, they come closer to optimally trading off between the competing pressures of simplicity and informativeness.

### 4.3. Discussion

We find a significant positive correlation between the degree of naturalness of an artificial language and how optimally a language trades off between the competing pressures of simplicity and informativeness. This suggests that efficient communication can explain the typology of quantifiers in natural language. One possible exception to this general trend concerns a cluster of languages in the bottom of Figure 2. (Thanks to an anonymous reviewer for pointing this out.) These languages all contain a single quantifier which does not belong to the set of quasi-natural ones, thus receiving degree of naturalness 0. At least two lessons can be drawn from the presence of this band of languages. On the one hand, it strengthens the main results: their zero degree and closeness to the Pareto frontier does not prevent the general correlation that we find. On the other hand, it does point to a limitation of our measure of degree of naturalness: it aggregates a *binary* property of individual expressions (namely, being quasi-natural or not), thus not being not very fine-grained. In particular, languages with a single quantifier either have degree 0 or degree 1. The measures introduced in the next section will not have this property.

## 5. Experiment 2: Degrees of Semantic Universals

The previous experiment showed that languages with more quasi-natural quantifiers tend to be more optimized for efficient communication. Now, the quasi-natural quantifiers are indeed monotone and conservative (after all, those are semantic universals), but they do not exhaust the space of possible quantifiers satisfying those properties. This raises the question: do these semantic universals on their own support efficient communication?

To address this question, we do two things: first (in the next subsection), we introduce information-theoretic measures of *degrees* of both monotonicity and conservativity. (A similar measure could also be made for quantity; see Section 3.1 where we note that quantity is assumed in the present work.) Then, instead of controlling how many quasi-natural quantifiers we sample, we randomly sample quantifiers from the space of all possible quantifiers to form languages and measure the resulting degrees of the universals for the languages, to test whether they are correlated with optimality or not.

### 5.1. Measuring Degrees of Universals

Both of our graded measures follow the same general recipe, described here. Both monotonicity and conservativity state that there is a *dependence* between the truth-values that a quantifier assigns to different models in the space of all models. For (upward) monotonicity: all ‘supermodels’ of a true model (those where the scope is widened) are also true. For conservativity: the ‘restriction’ of a true model must also be true (and similarly for falsehood), where the restriction of a model just replaces *B* with A∩B. For each property, we can turn the two types of models mentioned in the definitions into random variables, and then use (normalized) *mutual information* to measure the *amount of dependence* between them [21]. A fully monotone quantifier will have full dependence, but non-monotone quantifiers can have more or less such dependence (and *mutatis mutandis* for conservativity). We now make this intuition more precise.

#### 5.1.1. Monotonicity

According to the standard definition given above, monotonicity is a binary property. While one could compare monotone to non-monotone quantifiers in terms of their closeness to the Pareto frontier, some quantifiers are intuitively more monotone than other quantifiers. For instance, consider the three quantifiers “some”, “between 3 and 5” and “an even number of”. While “some” is monotone, and the other two quantifiers are not, intuitively, “an even number of” is the least monotone of the three because it is not equivalent to a Boolean combination of monotone quantifiers. Thus, we define a graded measure of monotonicity, which we can check for correlation with optimality. (See References [57,58], which show that this measure also increases over time during iterated learning.)

We measure upward monotonicity in information-theoretic terms as the proportion of uncertainty in the truth-value of a quantifier that is removed after knowing that there is a *submodel* where the quantifier is true. In particular, we define $M^{\prime }⪯M$ ($M^{\prime }$ is a submodel of M) just in case $M^{\prime }=M$, $A^{\prime }=A$, and $B^{\prime }\subseteq B$. For a perfectly upward monotone quantifier Q, if a structure M has a submodel which belongs to the quantifier, then M∈Q, as well. Therefore, for an upward monotone quantifier, all the uncertainty is removed, and the measure will have value 1.

More precisely, call M the set of all structures. Let {M,F,P} be a probability space with $F=2^{M}$, and *P* a discrete probability function with full support. (This assumption can be relaxed but is satisfied by the need probabilities (often uniform) used in efficient communication analyses.) Then, define two binary random variables ${𝟙}_{Q}$ and ${𝟙}_{Q}^{⪯}$ as follows, with M∈M:
$$\begin{array}{cc} {𝟙}_{Q}\left(M\right) & =1\;\mathrm{iff}\;M\in Q\\ {𝟙}_{Q}^{⪯}\left(M\right) & =1\mathrm{iff}\exists M^{\prime }⪯M\;s.t.\;M^{\prime }\in Q. \end{array}$$

In other words, ${𝟙}_{Q}$ is the indicator function of the quantifier Q. ${𝟙}_{Q}^{⪯}$ is 1 at a model just in case that model has a submodel that belongs to Q.

The entropy of ${𝟙}_{Q}$, $H({𝟙}_{Q})$, quantifies the uncertainty about what truth value Q will assign to a model. The conditional entropy $H({𝟙}_{Q}|{𝟙}_{Q}^{⪯})$ quantifies the uncertainty about what Q will assign to a model, given that one knows whether or not the model has a submodel belonging to it. An upward monotone quantifier minimizes (at value 0) $H({𝟙}_{Q}|{𝟙}_{Q}^{⪯})$: if one knows that a model has a true submodel, and the quantifier is upward monotone, one knows the truth value at that model. The difference between the entropy and the conditional entropy between these variables is known as the mutual information:
$$I\left({𝟙}_{Q};{𝟙}_{Q}^{⪯}\right):=H\left({𝟙}_{Q}\right)-H\left({𝟙}_{Q}|{𝟙}_{Q}^{⪯}\right).$$

This measures how much information ${𝟙}_{Q}^{⪯}$ provides about ${𝟙}_{Q}$. For an upward monotone quantifier, $H({𝟙}_{Q}|{𝟙}_{Q}^{⪯})=0$, so $I\left({𝟙}_{Q};{𝟙}_{Q}^{⪯}\right)=H\left({𝟙}_{Q}\right)$. In other words: for an upward monotone quantifier, knowing which structures have a true substructure provides as much information as knowing the entire quantifier.

While this roughly captures what we want from a measure of upward monotonicity, it needs to be normalized to form a degree that applies across quantifiers, since $0\leq I\left({𝟙}_{Q};{𝟙}_{Q}^{⪯}\right)\leq H\left({𝟙}_{Q}\right)$. We do this by dividing by $H({𝟙}_{Q})$, moving the upper bound to 1. (This is sometimes called the *uncertainty coefficient*, defined at least as early as Reference [59]). Overall, then, we measure upward monotonicity as
$$\begin{array}{cc} \mathrm{mon}(Q) & :=\frac{I({𝟙}_{Q};{𝟙}_{Q}^{⪯})}{H({𝟙}_{Q})}\\ & =\frac{H\left({𝟙}_{Q}\right)-H\left({𝟙}_{Q}|{𝟙}_{Q}^{⪯}\right)}{H({𝟙}_{Q})}\\ & =1-\frac{H({𝟙}_{Q}|{𝟙}_{Q}^{⪯})}{H({𝟙}_{Q}).} \end{array}$$

To see how this measure tracks intuitions, References [57,58] reports results for the previously mentioned quantifiers “some”, “between 3 and 5”, and “an even number of” on all models of a fixed size. “Some” gets monotonicity 1.0 because knowing whether a structure has a substructure that verifies “some” eliminates all uncertainty about the truth of the structure. The quantifier “between 3 and 5” has degree 0.7517, and the one with “an even number of” has degree 0.001, which captures the intuitive order of monotonicity of these quantifiers. We can also show that maximal degree behaves as desired.

**Proposition** **1.**

mon(Q)=1 iff Q

*is upward monotone.*


**Proof.** First, note that the degree is one just in case $H({𝟙}_{Q}|{𝟙}_{Q}^{⪯})=0$. It is commonly known that H(X|Y)=0 iff X=f(Y) for some function *f* from *Y*’s value space to *X*’s value space. In our case, this means that mon(Q)=1 iff there is an f:{0,1}→{0,1} such that ${𝟙}_{Q}\left(M\right)=f\left({𝟙}_{Q}^{⪯}\left(M\right)\right)$. Now, because ⪯ is a reflexive relation, ${𝟙}_{Q}^{⪯}\left(M\right)=0$ implies that ${𝟙}_{Q}\left(M\right)=0$ in general. Finally, note that we must have f(1)=1. Consider a model M such that ${𝟙}_{Q}^{⪯}\left(M\right)=1$. Then, for some $M^{\prime }⪯M,{𝟙}_{Q}\left(M^{\prime }\right)=1$. Because ⪯ is reflexive, we also have that ${𝟙}_{Q}^{⪯}\left(M^{\prime }\right)=1$, so f(1)≠0. The reader can verify that f(0)=0 and f(1)=1 just in case Q is upward monotone.    □

The measure defined above is a measure of upward monotonicity. It can be straightforwardly modified to measure downward monotonicity, by replacing the variable ${𝟙}_{Q}^{⪯}$ for the variable ${𝟙}_{Q}^{⪰}$, which is true when a model has a *supermodel* that belongs to Q. Similarly, these two measures have focused on *right* monotonicity, looking at truth-preservation in the nuclear scope of a quantifier. It is common to also consider monotonicity when moving to more or less general *restrictors*. The measure of monotonicity can also be applied to the restrictor by replacing *B* by *A* in the definitions of ⪯ and ⪰.

Finally, we take the measure of monotonicity of a single quantifier to be the average of its degrees of left and right monotonicity, where each of these is the *maximum* value of the respective degrees of upward and downward monotonicity (in the left versus right argument). In addition, we take the degree of monotonicity for a *language* to be the mean of the degrees of monotonicity of the quantifiers in the language.

#### 5.1.2. Conservativity

We define a graded measure of *degree of conservativity* in a similar way. It will still be a normalized mutual information, but now with a different random variable, capturing the fact that it suffices to look at A∩B instead of *B* for conservative quantifiers. More precisely, define the random variable
$${𝟙}_{Q}^{↾A}\left(M\right)=1\;\mathrm{iff}\;M↾A:=\langle M,A,A\cap B\rangle \in Q.$$

In other words: this variable returns 1 just in case the quantifier is true when restricting the scope by the restrictor. Finally, our measure of conservativity is:
$$\mathrm{cons}\left(Q\right):=\frac{I({𝟙}_{Q};{𝟙}_{Q}^{↾A})}{H({𝟙}_{Q})}=1-\frac{H({𝟙}_{Q}|{𝟙}_{Q}^{↾A})}{H({𝟙}_{Q})}$$

It is easy to show that maximal degree behaves as desired.

**Proposition** **2.**
*cons(Q)=1iffQ is conservative.*


**Proof.** This follows from the fact that Q is conservative iff ${𝟙}_{Q}={𝟙}_{Q}^{↾A}$.    □

Finally, we note that we can also define a measure of *left* conservativity by replacing the variable ${𝟙}_{Q}^{↾A}$ with the similarly defined
$${𝟙}_{Q}^{↾B}\left(M\right)=1\;\mathrm{iff}\;M↾B:=\langle M,A\cap B,B\rangle \in Q.$$

See Reference [60] and references therein for motivation for this form of conservativity. We take the degree of conservativity of a quantifier to be the *maximum* of its degree of left and right conservativity, and then the degree of conservativity of a language to be the average of the degrees of conservativity of the quantifiers therein.

### 5.2. Sampling Procedure

For this experiment, our sampling procedure was effectively random. For each number of words between 1 and 10, we generated 2000 languages containing that many words. The quantifiers in each language were chosen randomly from the set of all quantifiers whose minimal formula has 12 or fewer operators.

### 5.3. Results

The main results can be seen in Figure 3. The axes are the same as before, but with a different interpretation of the color of each language. In the left panel, color corresponds to degree of monotonicity; in the right panel, color corresponds to degree of conservativity, as defined in Section 5.1. We note that the horizontal ‘bands’ in this Figure (i.e., groups of languages with similar complexity) correspond roughly to the number of words in a language: the least complex languages have a single word with low complexity, while the most complex have many words with high complexity.

In these plots, it is not immediately clear whether the degrees of the universal properties are getting higher as the languages get closer to the estimated Pareto frontier. For degree of monotonicity, we find the Pearson correlation with optimality to be ρ=−0.0590 (boostrapped CI: [−0.07460891,−0.04257208]); for degree of conservativity, we find the same correlation to be 0.0725 (bootstrapped CI: [0.0565,0.0883]). These are incredibly weak correlations, which we interpret as showing that neither the degree of monotonicity nor the degree of conservativity are positively correlated with closeness to the Pareto frontier in this random sample of languages. (We note that these correlations are between an information-theoretic property of a language (the degrees), and their optimality (also measured using tools from information theory), but that these correlations do not directly reflect the entropy of a language nor the mutual information I(M;Q) discussed in Section 3.1. We thank an anonymous reviewer for noting this).

Although neither degree on its own is correlated with optimality, it is possible that the interaction between the two properties are, i.e., that languages that are *both* more monotone and more conservative are closer to the Pareto frontier. (Thank you to Jakub Szymanik (p.c.) for suggesting this analysis.) To test this, we measured the correlation between optimality and the product of the two degrees (of monotonicity and conservativity), finding ρ=0.0337 (bootstrapped CI: [0.0171,0.0500]).

### 5.4. Discussion

We introduced information-theoretic measures for *degrees* of monotonicity and conservativity, and we found that neither one of these degrees (nor their interaction) is significantly correlated with optimality in a random sample of languages. While this suggests that these properties on their own do not support efficient communication, some caution must be taken: Figure 3 shows that very few of the randomly sampled languages come close at all to the Pareto frontier. This suggests that the Pareto frontier lies in a very low-density region of the space of languages and that our method of randomly sampling did not effectively sample from that region. In addition, it is similarly possible that these low-density regions are also regions of high degrees of monotonicity and conservativity. Future work will explore methods for more exhaustively sampling from the vast space of possible languages.

## 6. General Discussion

Let us take stock. We have analyzed quantifiers in natural language from an efficient communication perspective, measuring how optimally quantifier systems trade-off between competing pressures for simplicity and informativeness. Experiment 1 showed that the degree of naturalness—the proportion of quasi-natural quantifiers in a language—is significantly correlated with optimality, suggesting that more natural languages are more optimal. Experiment 2 used information-theoretic measures of degrees of monotonicity and conservativity and found that, in a random sample of languages, neither of these degrees is significantly correlated with optimality. In this section, we discuss, in turn, (i) what these results mean for the status of semantic universals, (ii) how they are related to the linguistic laws literature, and (iii) what future work they suggest.

### 6.1. Status of Semantic Universals

Taken together, these two results suggest a strong conjecture: the semantic universals for quantifiers may be epiphenomena. That is to say, the fundamental pressure shaping quantifier systems cross-linguistically may be efficient communication; the fact that all natural language quantifiers are, for instance, monotone and conservative would be an empirically true by-product of these forces. On such a picture, then, there would not be an independent explanatory burden to explain why quantifiers satisfy the particular semantic universals (e.g., monotonicity) that they do, i.e., what makes monotone quantifiers special compared to non-monotone. Rather: a more general pressure shapes quantifier systems, and it so happens that the resulting systems have only monotone quantifiers.

What goes for quantifier systems may also hold for other semantic domains. For example, while color terms have been argued to be convex [16], and more convex systems are easier to learn [25], these may be epiphenomena of a pressure for efficient communication [30]. Similarly, while [27] shows that systems of indefinites satisfying proposed universals are closer to optimal than those that do not, this effect is significantly smaller than a comparison between natural versus artificial languages. This, again, suggests that the universals for indefinites are empirically true generalizations that arise as a by-product of a pressure for efficient communication.

It must be emphasized, however, that the two results of this paper merely suggest, but by no means force, this interpretation. As mentioned in Section 5.4, the lack of correlation between the degrees of monotonicity and conservativity and optimality may also be an artifact of the random sampling procedure employed. The most optimal languages appear to lie in low-density regions of the space of languages; these low-density regions, which were missed by our random sampling, may also be regions with high degrees of monotonicity and conservativity. Future work will develop sampling methods that do not directly sample natural language quantifiers (as in Experiment 1) but which still allow for sampling from regions of the space closer to the Pareto frontier than random sampling.

It must also be noted that the claim that natural languages are (nearly) Pareto-optimal does not necessarily imply that language change directly optimizes for these competing pressures. It possible that other general cognitive pressures which drive language change (e.g., the bottleneck in learning across generational transmission [61,62]) are explanatorily prior. Future work should directly integrate these measures of efficient communication with methods and models from language change.

### 6.2. Relationship to Linguistic Laws

While this paper has studied one set of semantic universals, another body of literature has studied large-scale statistical patterns in language, often under the name *linguistic laws* [9,10,11,12,13,14]. A celebrated example is Zipf’s [8] law, stating that the frequency and rank of word types in natural languages are related by a power law. See Reference [10] for a review of variants of this law and models thereof and Reference [63] for a thorough exploration of the connection between information theory and power laws in language. Similar laws have been identified in texts [11], speech [12,13], and in primate gestural communication [14]. In the subsequent paragraphs, I explain some similarities and differences, both in the nature of these laws and the universals discussed here, as well as in models for both.

Many linguistic laws concern properties of linguistic behavior that are *directly observable at a large scale*. This includes word frequencies, which are easily measurable from corpora, as well as speech data and deeper syntactic properties [5,6]. Large-scale measurement is essential to the formulation of the kind of statistical patterns characteristic of this tradition. By contrast, the semantic universals discussed here are generalizations about what types of meanings are expressed in languages, on the basis of a relatively small number of observations. One reason for this is that there are very few categories of expressions where there is enough consensus about their semantics to even formulate these universals. While it is hard to measure semantic phenomena at the kind of scale done in syntax (though, see Reference [64]), it would be a worthwhile pursuit to attempt to formulate and explain ‘semantic laws’ in a similar vein.

One popular explanation of the aforementioned linguistic laws comes from information theory [9,65]. In particular, where *R* is a random variable over possible meanings (e.g., objects of reference), and *S* a random variable over forms, they argue that languages optimize
Ω(λ):=−λI(S;R)+(1−λ)H(S),
where I(·;·) is mutual information, and H(·) is entropy [21]. For example, Reference [9] finds that optimal form-meaning mappings around λ≈.41 exhibit frequencies that pattern according to Zipf’s law. (See Reference [10] for an overview of this and other explanations of Zipf’s law.) Note that I(S;R) can be seen as a measure of the informativeness of a language, as discussed in Section 3.1. Similarly, one can argue that H(S) is a measure of the complexity of a language, so that minimizing Ω amounts to trying to jointly optimize pressures for simplicity and informativeness (see Reference [65] for a defense of the particular form that Ω takes). The information bottleneck (IB) approach [20,30] is a similar approach, with a different measure of complexity.

While the approach to communicative efficiency (i.e., a trade-off between simplicity and informativeness), thus, bears strong resemblance to this work on linguistic laws, there are at least two important divergences. On the one hand, both of our measures of informativeness and complexity exploit structure that is particular to the semantic domain in question, namely quantifiers. This is reflected in the utility measure $u(M^{\prime },M)$ and the particular language of thought used to measure complexity (Section 3.1). It remains to be seen whether domain-general measures of both can also explain the distribution of quantifiers in natural language, or whether such domain-specific concepts are necessary. Furthermore, Ω (and the corresponding equation in the IB approach) introduces a parameter λ for trading-off the two pressures, and then looks at minima for each λ. Our approach to optimization used a different principle: the evolutionary algorithm of Section 3.2 relies on the principle of *dominance avoidance*. A language is considered optimal if no other language is as good or better on both measures. We leave for future work an investigation of whether a simple functional form that combines the two measures in this paper and then minimizes would also work.

### 6.3. Future Work

There are other directions for future work that could effect the results and the interpretation thereof. In the present paper, we assumed uniform need probabilities over the space of possible models when measuring informativeness. Ideally, such need probabilities could be better estimated empirically (e.g., from corpora [18,27]). Because, however, the semantic space here is quite abstract, developing tools for this estimation is a non-trivial task. Furthermore, the present paper assumed the semantic universal of Quantity because relaxing this assumption causes an exponential increase in the size of the semantic space (see Section 3.1). A *degree of quantity* can be defined similarly to our measures for monotonicity and conservativity. It is possible that this degree, and the other two, as well, would be correlated with optimality in the richer semantic space. Future work can pursue computationally efficient methods for carrying out these analyses in the face of the aforementioned combinatorial explosion. In that vein, Reference [66] recently measured simplicity in the same manner as here, but without assuming Quantity, showing that quantitative, conservative, and monotone quantifiers are all simpler than ones without those properties. Thus, it may also be possible that quantifiers are a domain where simplicity on its own suffices, without appeal to informativeness. Finally, the idea behind the degrees of semantic universals in this paper can also be applied to other linguistic domains; doing so can help clarify the generality of the results in this paper.

In addition to these variations on the experiments reported in this paper, additional empirical work on the quantifier systems of the languages of the world could help support the claim that these systems are optimized for efficient communication. In domains, such as kinship [18], color [20], and indefinites [27], earlier typological work has provided a robust accounting of many languages’ inventories in those domains. As mentioned in Section 4.1, while there have been cross-linguistic investigations concerning quantifiers [54,55,67], these have not yielded a precise account of exactly which quantifiers are expressed by simple determiners in a large group of the world’s languages. Such a resource would allow us to compare more directly whether quantifiers in natural language support efficient communication, instead of relying on alternative measures of naturalness as in the present study.

## 7. Conclusions

In conclusion, we have conducted an efficient communication analysis of quantifiers in natural language, using information-theoretic measures of degrees of semantic universals. These results suggest, but do not entail, that semantic universals may not stand in need of independent explanation but, rather, arise as epiphenomena of the trade-off between simplicity and informativeness. Future work will refine these results and their interpretation, as well as extend the application of information-theoretic degrees of universals to other semantic domains.

## Figures and Tables

**Figure 1 entropy-23-01335-f001:**
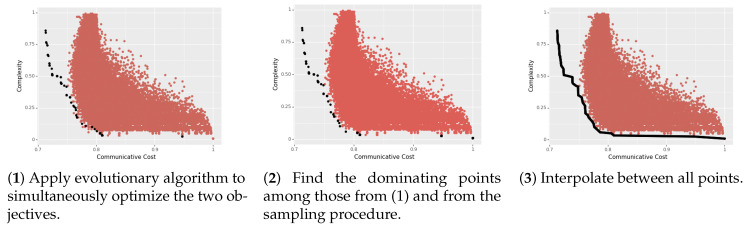
The overall Pareto frontier estimation algorithm, in three steps. Each red point is one artificial language sampled according to an independent procedure. The black points in panel (3) constitute the final estimate of the true Pareto frontier.

**Figure 2 entropy-23-01335-f002:**
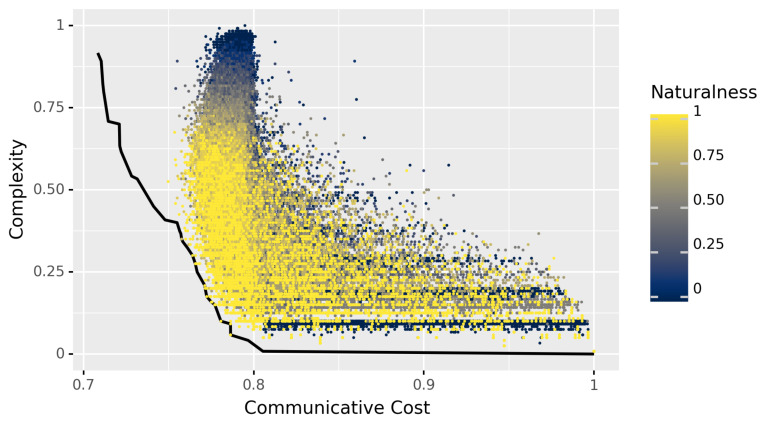
Languages in the space of communicative cost and complexity (see Section 3.1, with cost defined as 1−I(L)), colored by their degree of naturalness. Languages with more quasi-natural quantifiers appear to be closer to optimal, as measured by closeness to the (estimated) Pareto frontier, depicted in black.

**Figure 3 entropy-23-01335-f003:**
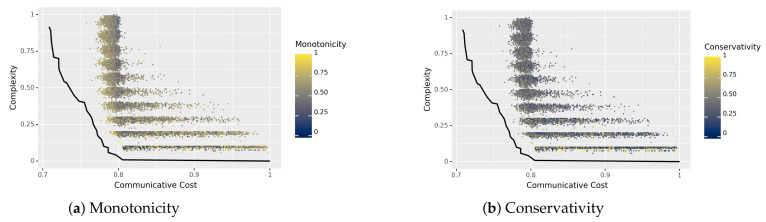
Languages in the space of communicative cost and complexity, colored by their degree of (**a**) monotonicity and (**b**) conservativity. Neither degree correlates with optimality, as measured by closeness to the (estimated) Pareto frontier, depicted in black.

**Table 1 entropy-23-01335-t001:** The operators in the grammar for generating quantifiers.

Boolean	Set-Theoretic	Numeric
∧, ∨, ¬	∩,∪,⊂,|·|	/,+,−,>,=,%

## Data Availability

The data generated in both experiments can be found at the following repository on the Open Science Framework: https://doi.org/10.17605/OSF.IO/Y58K4 (accessed on 9 October 2021). The code for generating and analyzing the data may be found at the following GitHub repository (also linked from OSF): https://github.com/shanest/SimInf_Quantifiers (accessed on 9 October 2021).

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
