# Peer review of "Quantifiers in Natural Language: Efficient Communication and Degrees of Semantic Universals"

_entropy, 2021, doi:10.3390/e23101335_

Round 1

Reviewer 1 Report

Review :  

Quantifiers in Natural Language: Efficient Communication and

Degrees of Semantic Universals

In this article, the author have tried to study the presence of semantic universals in a domain of function words (quantifiers), considering its potential relationship with linguistic universals. The work is interesting and I congratulate the author for it, but nevertheless it still needs some revision for its acceptance in Entropy.

As it is written now, the paper needs revision in theoretical aspects and in some technical details, where the level of deepening is insufficient for Entropy standards. In this sense I recommend it for publication, but only when the main issues listed below are solved.

Main Issues

Abstract

0.- Please improve this ambiguous (and empty) definition “Semantic universals are properties of meaning that are shared across languages.” What is the definition of semantic universals used here? In my opinion, one option is to use the one given in the introduction (lines 15-16). In line 6 author claim: “more natural languages are more optimal”, What does 'natural language' mean? Is it compared in the article with artificial languages? Please clarify this. In line 77 it is said: “In what follows, a language is a set of quantifiers.” So here author study ‘sets of quantifiers’, not natural language?

Introduction

1.- The introduction begins with a general paragraph that should be completed with some basic references about semantics. In lines 15-16 in this definition: “Semantic universals are restrictions on the range of variation in meaning across languages.” a general reference should be added, and I think that Entropy readers need some details/discussion about “semantic universals”. What are that ‘restrictions of meaning’? Are they related to frequency (Zipf’s law?)? Please explain more about this topic in introduction. The same could be done about “degree of naturalness”.

2.- In lines 19-25, the author raises ('roughly') a not trivial problem of optimization of linguistic systems that has been discussed previously from the perspective of information theory (see recently Debowski (2020), Gibson et al.(2019) or Cover and Thomas (2006)) and ‘sender-receiver’ models. Linguistic optimization has been related to both power laws and efficiency from the perspective of the sender or receiver. That is, traditionally it is said that if only one word is used for all meanings that would be communicatively unfeasible, while using one word for each meaning is not “economical”…and then a balance emerge and Zipf’s law (Ferrer-i-Cancho and Solé, 2003; Ferrer-i-Cancho, 2005) . What is your explanation about that ‘sender-receiver’ balance and Zipf’s law (determiners has different frequency/rank in language) as a consequence of this balance? Is there any connection between power laws (Piantadosi, 2014; Ferrer-i-Cancho and Solé, 2003; Ferrer-i-Cancho, 2005) or other statistic/linguistic laws (Torre et al, 2019) with the findings and statements presented here about optimality in language (see i.e. Ferrer Cancho, 2018; Gibson et al. 2019)?

3.- Finally, I think that Notes on terminology (see below, point 5) should be introduced in this first section.

  1. Background

4.- The article is not academic in the background. Classical references on basic definitions and properties are required (lines 64-67). As a suggestion, a table or diagram would help in the presentation of the general definitions and examples of the lines 64-69. Are simple determiners = single words and complex determiners = more than one word? Please clarify this.

5.- Some comments about the issue of study units are necessary. Lines 70-77 should be revised and rewritten in a way that is more understandable to the multidisciplinary Entropy reader. What is Q? What is the structure/model M? What is the difference between quantifier and determiner? It is hard to follow this article if these definitions are not clear. The author refers to another reference [6] in a footnote. I think (s)he should make an explanatory effort here. On the other hand, the notes on terminology should go before its use in previous lines.

6.- Section 2.2. should start with a short explanatory paragraph of what comes next. I like section 2.2. but there should be a better explanation of the terminology beforehand, as I said before in 5.

7.- About the monotonicity universal “All simple determiners are monotone” (line 103).  Does it always happen with simple determiners that are a single word? Are there cases of simple determiners that are not a single word? The same about CONSERVATIVITY UNIVERSAL: All simple determiners are conservative. (line 121). Can you give examples in complex determiners or they don’t exist? Please clarify this.

8.- On the learnability hypothesis, can the author argue that the point is not that it is easier for children to learn a simple word than a set of words? (if really the simple determiners -monotone, conservative- are a single word). The quantitative argument seems to me simpler than the one offered by the author.

9.- Gibson et al. (2019) define:  Efficiency: a code is efficient if successful communication can be achieved with minimal effort on average by the sender and receiver, usually by minimizing the message length.

And in Lines 154-157, author claim again: “A maximally simple language may have a single word; this would be very unhelpful for communication, since it does not allow speakers to  make any distinctions. By contrast, a maximally informative language may have a unique expression for every possible meaning; this would be, however, very complex to represent or learn”

The point is, How does the author account for this ‘efficiency’ balance (I suppose (s)he agree with Gibson’s definition)? Seminal works by Zipf are missed, or more recently Ferrer-i-Cancho or Debowski (2020) explore that distinction, via a cost function (see Ferrer i Cancho and Solé (2003)). On the other hand, theory about brevity (Zipf’s law of brevity) and polysemy (Zipf’s Meaning-frequency law) in language should be mentioned here or/and in discussion, properly. 

10.- It is essential to define the concepts of cognitive and communicative cost in a clear way. What is the difference between them? I don’t understand dots in figure 1. Caption should be more understandable to a broad audience.

11.- Finally, author clarify (lines 171-173) that “the present paper will compare not just natural versus artificial languages, but will also compare random languages that do and do not satisfy proposed semantic universals for the domain in question (indefinites there, quantifiers here).” This should be explained in abstract and in the beginning of the paper more clearly.

  1. Methods

  1. Previous comment: In this section I find some elements of the introduction (definition of concepts) and some results (eg: figure 2). It confuses a bit, I think the author should review it especially. Minor comment: Footnotes 7 and 8 are important (in line 313 footnote 7 appear again…) and should go in the main text.

  1. Author consider (lines 188-189): “The complexity of a quantifier is the length of the shortest formula in this language that denotes the quantifier”. So, Wouldn't it be expected to recover some analogy to the so-called law of brevity (Zipf’s law of abbreviation, see Torre et al.(2019))? In other words, does the frequency of use of each symbolic element influence the results?

14.- Lines 198-202: This explanation is very enlightening and useful and I strongly recommend placing it in the introduction (see comment 5). A figure could be very useful too.

15.- On the Pareto frontier, a key point of the article, is there any relationship with the Zipfian approach? Has no function been able to fit the distribution of points of the Pareto frontier (figure 2)? this analysis would increase the impact of the article.

16.- Ferrer-i-Cancho (2004) explores the Euclidean distance minimization in syntax. Is there any correspondence between syntax and semantics under this minimization approach?

17.- I find the concept of 'mutation' raised very interesting, and in this sense it would be well to make explicit what the total combinatorial potential is, given the number of symbols used (lines 233-238).

4-5. Experiments/Results

18.- The results subsection 4.2 is frankly interesting, but a clear explanation of the communicative cost and complexity computation is missing here. How are they calculated (figure 3, improve caption in this regard)? It is not explained well and, in another way, it gives the feeling of a circular argument: is a language more “natural” the closer it gets to the Pareto frontier, or Languages with more quasi-natural quantifiers appear to be closer to optimal Pareto frontier? Come black line/Pareto frontier in fig. 3 from fig. 2? Please clarify this subsection 4.2.

19.- Author claim (lines 290-292): “We find a significant positive correlation between the degree of naturalness of an artificial language and how optimally a language trades off between the competing pressures of simplicity and informativeness”. Is simplicity equivalent to ‘brevity’ or ‘efficiency’ of coding?

20.- Line 326: Please include the formula/definition of mutual information and entropy (I suppose as a classical reader that it is Shannon Entropy) used here, and the corresponding references (Cover and Thomas, 2006; Shannon 1948?).

21.- Upward monotonicity is proportional to mutual information; Ferrer-i-Cancho (2018) introduce two information theoretic principles: maximization of mutual information between forms and meanings and minimization of form entropy. I think it is necessary to relate in discussion the results presented here with this previous work and the approach of these two principles of language optimization.

22.- Author claim that (lines 403-406): “These are incredibly weak correlations, which we interpret as showing that neither the degree of monotonicity nor the degree of conservativity are positively correlated with closeness to  the Pareto frontier in this random sample of languages.” Information theory is used but it is not discussed how this 'incredibly weak' relationship relates to mutual information or entropy of language. This is a very important topic to review for Entropy readers.

23.- The “bands” that appear in figure 4 dots are due to technical reasons that should be explained, in the main text and in the caption.

6.- General discussion/Conclusion

24.- In the general discussion the author seems to only take into account his/her results in the study of semantics / quantifiers. In my opinion, other crucial elements in communication are being overlooked, such as syntax or approaches to the statistical patterns of language. What’s your opinion about the principle of compression (Ferrer Cancho et al, 2013; Cover and Thomas, 2006) or brevity law (the tendency of more frequent elements to be shorter)? Can be related with these results? A paragraph about that in general discussion would be very interesting.

25.- Related to the previous point, what cognitive aspects does the author believe are behind these semantic optimization processes? In the general discussion this is a good time to speculate a bit more about the implications of his work.

26.- As a future work it seems that at no time does the author consider analyzing what happens with real languages, studying their quantifiers, etc. The work seems interesting to me, but I think that it is necessary to differentiate the models of the reality that it is intended to study and to raise at the end the need to take the step to the analysis of real languages.

27.- Finally, I strongly recommend author to review Debowski (2020) to relate their (future) work with a broad audience in linguistics/information theory community.

REFS:

Debowski, L. (2020). Information Theory Meets Power Laws: Stochastic Processes and Language Models. John Wiley & Sons.

Gibson, E., Futrell, R., Piantadosi, S. P., Dautriche, I., Mahowald, K., Bergen, L., & Levy, R. (2019). How efficiency shapes human language. Trends in cognitive sciences, 23(5), 389-407.

Cover, T. M. & Thomas, J. A. (2006). Elements of information theory, 2nd edition. Hoboken, NJ: Wiley.

Ferrer-i-Cancho, R. & Solé, R. V. (2003). Least effort and the origins of scaling in human language. Proceedings of the National Academy of Sciences USA, 100, 788-791

Ferrer-i-Cancho, R. (2005). Zipf's law from a communicative phase transition. The European Physical Journal B, 47(3), 449-457.

Torre, I. G., Luque, B., Lacasa, L., Kello, C. T., & Hernández-Fernández, A. (2019). On the physical origin of linguistic laws and lognormality in speech. Royal Society open science, 6(8), 191023.

Ferrer-i-Cancho, R. (2018). Optimization models of natural communication. Journal of Quantitative Linguistics, 25(3), 207-237.

Piantadosi, S. T. (2014). Zipf’s word frequency law in natural language: A critical review and future directions. Psychonomic bulletin & review, 21(5), 1112-1130.

Ferrer-i-Cancho, R. (2004). Euclidean distance between syntactically linked words. Physical Review E, 70(5), 056135.

Ferrer-i-Cancho, R., Hernández‐Fernández, A., Lusseau, D., Agoramoorthy, G., Hsu, M. J., & Semple, S. (2013). Compression as a universal principle of animal behavior. Cognitive Science, 37(8), 1565-1578.

Author Response

> Review: Quantifiers in Natural Language: Efficient Communication and Degrees of Semantic Universals

> In this article, the author have tried to study the presence of semantic universals in a domain of function words (quantifiers), considering its potential relationship with linguistic universals. The work is interesting and I congratulate the author for it, but nevertheless it still needs some revision for its acceptance in Entropy.

> As it is written now, the paper needs revision in theoretical aspects and in some technical details, where the level of deepening is insufficient for Entropy standards. In this sense I recommend it for publication, but only when the main issues listed below are solved.

I thank the reviewer both for their kind words here and for the extremely helpful suggestions made throughout a close read of the paper.  My responses to particular points raised are in-line below.  The paper is certainly much improved thanks to this.

> Main Issues

> Abstract

> 0.- Please improve this ambiguous (and empty) definition “Semantic universals are properties of meaning that are shared across languages.” What is the definition of semantic universals used here? In my opinion, one option is to use the one given in the introduction (lines 15-16). In line 6 author claim: “more natural languages are more optimal”, What does 'natural language' mean? Is it compared in the article with artificial languages? Please clarify this. In line 77 it is said: “In what follows, a language is a set of quantifiers.” So here author study ‘sets of quantifiers’, not natural language?

Thank you; I have made the definition of semantic universals more precise and reworded the claim about 'more natural languages'.  I hope that these clarify the content of the paper and are easier to follow.

> Introduction

> 1.- The introduction begins with a general paragraph that should be completed with some basic references about semantics. In lines 15-16 in this definition: “Semantic universals are restrictions on the range of variation in meaning across languages.” a general reference should be added, and I think that Entropy readers need some details/discussion about “semantic universals”. What are that ‘restrictions of meaning’? Are they related to frequency (Zipf’s law?)? Please explain more about this topic in introduction. The same could be done about “degree of naturalness”.

I have added an example of a semantic universal in the introduction to make it clearer (in addition to providing examples of other linguistic universals in response to Reviwer 2).  Degree of naturalness is defined in the introduction, but I have re-worded it slightly to make it clearer that this is a new term / concept.

> 2.- In lines 19-25, the author raises ('roughly') a not trivial problem of optimization of linguistic systems that has been discussed previously from the perspective of information theory (see recently Debowski (2020), Gibson et al.(2019) or Cover and Thomas (2006)) and ‘sender-receiver’ models. Linguistic optimization has been related to both power laws and efficiency from the perspective of the sender or receiver. That is, traditionally it is said that if only one word is used for all meanings that would be communicatively unfeasible, while using one word for each meaning is not “economical”…and then a balance emerge and Zipf’s law (Ferrer-i-Cancho and Solé, 2003; Ferrer-i-Cancho, 2005) . What is your explanation about that ‘sender-receiver’ balance and Zipf’s law (determiners has different frequency/rank in language) as a consequence of this balance? Is there any connection between power laws (Piantadosi, 2014; Ferrer-i-Cancho and Solé, 2003; Ferrer-i-Cancho, 2005) or other statistic/linguistic laws (Torre et al, 2019) with the findings and statements presented here about optimality in language (see i.e. Ferrer Cancho, 2018; Gibson et al. 2019)?

Thank you for pressing about the connection to the linguistic laws literature (as did Reviewer 2).  I have added references to the information theory literature in the introduction, and included a footnote about linguistic laws, pointing to a longer discussion about them in the general discussion section.

> 3.- Finally, I think that Notes on terminology (see below, point 5) should be introduced in this first section.

I found it difficult to include all of the terminology in the introduction, preferring to stay more informal here and focus on the main story and then getting precise in the next section.  If the reviewer strongly prefers, I can work on introducing all of the terminology this early.

> Background

> 4.- The article is not academic in the background. Classical references on basic definitions and properties are required (lines 64-67). As a suggestion, a table or diagram would help in the presentation of the general definitions and examples of the lines 64-69. Are simple determiners = single words and complex determiners = more than one word? Please clarify this.

Thank you for pointing this out.  References have aso been added to this paragraph and in a footnote.  I have added more information about the simple vs complex distinction, which does roughly map on to single vs multiple words.   

> 5.- Some comments about the issue of study units are necessary. Lines 70-77 should be revised and rewritten in a way that is more understandable to the multidisciplinary Entropy reader. What is Q? What is the structure/model M? What is the difference between quantifier and determiner? It is hard to follow this article if these definitions are not clear. The author refers to another reference [6] in a footnote. I think (s)he should make an explanatory effort here. On the other hand, the notes on terminology should go before its use in previous lines.

Thank you for pointing out the terseness of this section, especially for the intended audience of this paper.  I have expanded Section 2.1, including more details about simple vs. complex, what a model/structure is, and how determiners (syntactic units in a natural language) are different from quantifiers (the mathematical objects that are the meaning of determiners).  I have also added a quick example using the sentence "Every person is happy", which I hope will also clarify things.

> 6.- Section 2.2. should start with a short explanatory paragraph of what comes next. I like section 2.2. but there should be a better explanation of the terminology beforehand, as I said before in 5.

Thank you; such an introduction paragraph has now been added.

> 7.- About the monotonicity universal “All simple determiners are monotone” (line 103).  Does it always happen with simple determiners that are a single word? Are there cases of simple determiners that are not a single word? The same about CONSERVATIVITY UNIVERSAL: All simple determiners are conservative. (line 121). Can you give examples in complex determiners or they don’t exist? Please clarify this.

Indeed, simple roughly means single-word.  The expanded Section 2.1 (see above) makes this more explicit.  At the end of Section 2.2.1, I mention the complex determiner "at least 6 or at most 2" as an example that's non-monotonic.  For conservative determiners, many authors believe that _all_ NL determiners, simple and complex, are in fact conservative, and that this might be for complicated reasons about the syntax-semantics interface.  Footnote 10 comments on this.

> 8.- On the learnability hypothesis, can the author argue that the point is not that it is easier for children to learn a simple word than a set of words? (if really the simple determiners -monotone, conservative- are a single word). The quantitative argument seems to me simpler than the one offered by the author.

Here the question is not about learning a single word versus a set of words.  Rather, it's _while learning the meaning of a single word_, which ones are easier and harder?  The claim (at the end of the first paragraph in Section 2.3) is that languages assign the easy-to-learn meanings to single words, and this fact is independent of how many words the language has in general.

> 9.- Gibson et al. (2019) define:  Efficiency: a code is efficient if successful communication can be achieved with minimal effort on average by the sender and receiver, usually by minimizing the message length.

> And in Lines 154-157, author claim again: “A maximally simple language may have a single word; this would be very unhelpful for communication, since it does not allow speakers to  make any distinctions. By contrast, a maximally informative language may have a unique expression for every possible meaning; this would be, however, very complex to represent or learn”

> The point is, How does the author account for this ‘efficiency’ balance (I suppose (s)he agree with Gibson’s definition)? Seminal works by Zipf are missed, or more recently Ferrer-i-Cancho or Debowski (2020) explore that distinction, via a cost function (see Ferrer i Cancho and Solé (2003)). On the other hand, theory about brevity (Zipf’s law of brevity) and polysemy (Zipf’s Meaning-frequency law) in language should be mentioned here or/and in discussion, properly. 

I have made a footnote here in this section with some references, and also included more information in the general discussion.

> 10.- It is essential to define the concepts of cognitive and communicative cost in a clear way. What is the difference between them? I don’t understand dots in figure 1. Caption should be more understandable to a broad audience.

I have added a sentence providing more detail about how I'm thinking about these terms here, with a pointer to the fact that more precise definitions are coming in the next section.  I have also deleted Figure 1, because it seemed to cause more confusion that clarification amongst readers.

> 11.- Finally, author clarify (lines 171-173) that “the present paper will compare not just natural versus artificial languages, but will also compare random languages that do and do not satisfy proposed semantic universals for the domain in question (indefinites there, quantifiers here).” This should be explained in abstract and in the beginning of the paper more clearly.

Thank you; I have added a sentence in the abstract and the end of the introduction making this more explicit.

> Methods

> Previous comment: In this section I find some elements of the introduction (definition of concepts) and some results (eg: figure 2). It confuses a bit, I think the author should review it especially. Minor comment: Footnotes 7 and 8 are important (in line 313 footnote 7 appear again…) and should go in the main text.

I have made it clearer in the text and caption that Figure 2 is not a results figure, but a methodological one, in that it's how we estimate the Pareto frontier.  I have also moved footnotes 7 and 8 into the body of the text.

> Author consider (lines 188-189): “The complexity of a quantifier is the length of the shortest formula in this language that denotes the quantifier”. So, Wouldn't it be expected to recover some analogy to the so-called law of brevity (Zipf’s law of abbreviation, see Torre et al.(2019))? In other words, does the frequency of use of each symbolic element influence the results?

Note here that the formula here are meant to correspond to "mental representations", not actual linguistic elements, so they are never directly used in e.g. speech production.  That being said, it is possible to change the probabilities in the PCFG in a way that would make some symbolic elements more costly than others.  Intuitively, this corresponds to mental difficulty however, and not any notion of frequency of use. Where I have moved footnote 8 in to the body of the text, I have also clarified this point.

> 14.- Lines 198-202: This explanation is very enlightening and useful and I strongly recommend placing it in the introduction (see comment 5). A figure could be very useful too.

Thank you; I have added a clause in the introduction making the communicative scenario more explicit.

> 15.- On the Pareto frontier, a key point of the article, is there any relationship with the Zipfian approach? Has no function been able to fit the distribution of points of the Pareto frontier (figure 2)? this analysis would increase the impact of the article.

I have in the past also fit a variety of functions to the raw Parteo frontier data, but have not included it for the reason that most of them leave some of the sampled languages on the "other side" of the frontier, which should not in principle be possible.  If the author has any suggestions for how to prevent that, I'd welcome them.

> 16.- Ferrer-i-Cancho (2004) explores the Euclidean distance minimization in syntax. Is there any correspondence between syntax and semantics under this minimization approach?

No, there is in fact no explicit syntax on the appraoch in this paper.  I have, however, added a reference to Ferrer-i-Cancho 2004 in the introduction with a mention of dependency length minimization as a syntactic universal.

>17.- I find the concept of 'mutation' raised very interesting, and in this sense it would be well to make explicit what the total combinatorial potential is, given the number of symbols used (lines 233-238).

Thank you for asking for this.  I have noted now in Section 3.1 that the method for generating quantifier meanings resulted in 279120 distinct quantifiers.  I have also added a footnote in Section 3.2 that counts the number of possible languages, to show how very quickly it grows.

> 4-5. Experiments/Results

> 18.- The results subsection 4.2 is frankly interesting, but a clear explanation of the communicative cost and complexity computation is missing here. How are they calculated (figure 3, improve caption in this regard)? It is not explained well and, in another way, it gives the feeling of a circular argument: is a language more “natural” the closer it gets to the Pareto frontier, or Languages with more quasi-natural quantifiers appear to be closer to optimal Pareto frontier? Come black line/Pareto frontier in fig. 3 from fig. 2? Please clarify this subsection 4.2.

I have clarified in the main text and the caption that the measures are those taken from Section 3.1 (with a 1-informativeness as cost).

I am unsure about the nature of the alleged circular argument: in this paper, "more natural" just means "having more quasi-natural quantifiers", so those wind up being the same thing.  I hope that the earlier clarifications in the text about degree of naturalness help here as well.

> 19.- Author claim (lines 290-292): “We find a significant positive correlation between the degree of naturalness of an artificial language and how optimally a language trades off between the competing pressures of simplicity and informativeness”. Is simplicity equivalent to ‘brevity’ or ‘efficiency’ of coding?

Simplicity is as defined in Section 3.1, in terms of minimum expression length in a language of thought.  The connection with brevity is examined in the new part of the general discussion.

> 20.- Line 326: Please include the formula/definition of mutual information and entropy (I suppose as a classical reader that it is Shannon Entropy) used here, and the corresponding references (Cover and Thomas, 2006; Shannon 1948?).

I have added the reference here, but not the formula.  This is explicitly given in section 5.1.1, where it is also explained in more detail.  Mutual information is also explicitly defined in Section 3.1 of this revised version. Line 326 is just a transition paragraph to explain what comes next, so it seems better to leave the math to 5.1.1.

> 21.- Upward monotonicity is proportional to mutual information; Ferrer-i-Cancho (2018) introduce two information theoretic principles: maximization of mutual information between forms and meanings and minimization of form entropy. I think it is necessary to relate in discussion the results presented here with this previous work and the approach of these two principles of language optimization.

Very much agreed.  I have included a new subsection in the discussion that does this.

> 22.- Author claim that (lines 403-406): “These are incredibly weak correlations, which we interpret as showing that neither the degree of monotonicity nor the degree of conservativity are positively correlated with closeness to  the Pareto frontier in this random sample of languages.” Information theory is used but it is not discussed how this 'incredibly weak' relationship relates to mutual information or entropy of language. This is a very important topic to review for Entropy readers.

Thank you for noting this; I have added a footnote in Section 5.3 noting that the correlations are _between_ information-theoretic properties of a language, and so do not directly mirror entropy or mutual information.

> 23.- The “bands” that appear in figure 4 dots are due to technical reasons that should be explained, in the main text and in the caption.

Done.

> 6.- General discussion/Conclusion

> 24.- In the general discussion the author seems to only take into account his/her results in the study of semantics / quantifiers. In my opinion, other crucial elements in communication are being overlooked, such as syntax or approaches to the statistical patterns of language. What’s your opinion about the principle of compression (Ferrer Cancho et al, 2013; Cover and Thomas, 2006) or brevity law (the tendency of more frequent elements to be shorter)? Can be related with these results? A paragraph about that in general discussion would be very interesting.

This is a very fair point that reflects that this conclusion grew out of an earlier paper aimed at a semantics audience, not the Entropy readership.  I have re-structured the conclusion heavily and included a subsection addressing the points that you ask about here.

> 25.- Related to the previous point, what cognitive aspects does the author believe are behind these semantic optimization processes? In the general discussion this is a good time to speculate a bit more about the implications of his work.

I have added a paragraph to (the new) subsection 6.1 suggesting that a bottleneck in generational transmission could be a factor, and calling for more integration with models of language change.

> 26.- As a future work it seems that at no time does the author consider analyzing what happens with real languages, studying their quantifiers, etc. The work seems interesting to me, but I think that it is necessary to differentiate the models of the reality that it is intended to study and to raise at the end the need to take the step to the analysis of real languages.

This is a very good point, and I have added some discussion of this kind of work in the new future work subsection.

> 27.- Finally, I strongly recommend author to review Debowski (2020) to relate their (future) work with a broad audience in linguistics/information theory community.

Thank you, I have added a reference to this work as well.

REFS:

Debowski, L. (2020). Information Theory Meets Power Laws: Stochastic Processes and Language Models. John Wiley & Sons.

Gibson, E., Futrell, R., Piantadosi, S. P., Dautriche, I., Mahowald, K., Bergen, L., & Levy, R. (2019). How efficiency shapes human language. Trends in cognitive sciences, 23(5), 389-407.

Cover, T. M. & Thomas, J. A. (2006). Elements of information theory, 2nd edition. Hoboken, NJ: Wiley.

Ferrer-i-Cancho, R. & Solé, R. V. (2003). Least effort and the origins of scaling in human language. Proceedings of the National Academy of Sciences USA, 100, 788-791

Ferrer-i-Cancho, R. (2005). Zipf's law from a communicative phase transition. The European Physical Journal B, 47(3), 449-457.

Torre, I. G., Luque, B., Lacasa, L., Kello, C. T., & Hernández-Fernández, A. (2019). On the physical origin of linguistic laws and lognormality in speech. Royal Society open science, 6(8), 191023.

Ferrer-i-Cancho, R. (2018). Optimization models of natural communication. Journal of Quantitative Linguistics, 25(3), 207-237.

Piantadosi, S. T. (2014). Zipf’s word frequency law in natural language: A critical review and future directions. Psychonomic bulletin & review, 21(5), 1112-1130.

Ferrer-i-Cancho, R. (2004). Euclidean distance between syntactically linked words. Physical Review E, 70(5), 056135.

Ferrer-i-Cancho, R., Hernández‐Fernández, A., Lusseau, D., Agoramoorthy, G., Hsu, M. J., & Semple, S. (2013). Compression as a universal principle of animal behavior. Cognitive Science, 37(8), 1565-1578.

Reviewer 2 Report

In the paper Quantifiers in Natural Language: Efficient Communication and Degrees of Semantic Universals the author analysis the language universals on quantifier words. For doing this they use a map languages on a set of experiments comparing complexity and comunicative cost. Their results suggest that both that efficient communication shapes semantic typology in both content and that semantic universals may not stand in need of independent

first of all I would like to congratulate the author for this interesting article. Some parts of the article I have not reviewed in depth due to lack of sufficient knowledge of the specialty. However, I have some questions that I would like to ask.

Introduction

Probably the introduction could be improved, considering more explanations and references. Particularly:

  • “While the languages of the world vary greatly, linguists have discovered many restrictions on possible variation”. What restrictions? Could you put some examples? Are physiological constrains considered here (density of the air, size of lunges, etc)?

    • Lines 20-26. Good explanation. I suggest to cite Elements of Information theory” from Joy A. Thomas y Thomas M. Cover. The concept of entropy could also be introduced.
    • “The general claim: the semantic systems of the world’s languages optimally balance these two competing pressure” You could probably refer to this paper “Least effort and the origins of scaling in human language” Cancho and Sole; where it is shown that languages should lie close to a phase transition.

    • Other universals could be introduced, for example recently. “Hahn, M., Jurafsky, D., & Futrell, R. (2020). Universals of word order reflect optimization of grammars for efficient communication. Proceedings of the National Academy of Sciences117(5), 2347-2353.” .
    • Linguistic laws are also universals (probably those that have been quantitatively measured in a more massive way), where probably the most popular is Zipf-law which has been found in all languages and predicted in “Zipf, G. K. (2016). Human behavior and the principle of least effort: An introduction to human ecology. Ravenio Books.” Those linguistic laws have been further found and studied in texts (Altmann, E. G., & Gerlach, M. (2016). Statistical laws in linguistics. In Creativity and universality in language (pp. 7-26). Springer, Cham.), speech (Torre, I. G., Luque, B., Lacasa, L., Kello, C. T., & Hernández-Fernández, A. (2019). On the physical origin of linguistic laws and lognormality in speech. Royal Society open science6(8), 191023.), in microscopic scales of human voice (Torre, I. G., Luque, B., Lacasa, L., Luque, J., & Hernández-Fernández, A. (2017). Emergence of linguistic laws in human voice. Scientific reports7(1), 1-10.) and even animal communication (Heesen, R., Hobaiter, C., Ferrer-i-Cancho, R., & Semple, S. (2019). Linguistic laws in chimpanzee gestural communication. Proceedings of the Royal Society B286(1896), 20182900.).

    • Lines 151-160. References are needed. This paragraph could be explained in terms of Principle of least effort (Zipf) or similarly in terms of Information theory (Cover Thomas or Shannon).
    • “Pareto frontier”. The search of Pareto optimal solutions should be better introduced.
    • It would be interesting to plot optimality versus Degree of naturalness to check if there is indeed a linear correlation or maybe some other type of function between those variables. The fulfillment of Pearson correlation hypothesis should also be checked (lines 283-285).
    • Degree of monotonicity: As far as this reviewer known, there is not some kind of degree of monotonicity and from the mathematical point of view it would make no sense at all. Maybe another term should be chosen or scientific references should be reported to backward the term “degree of monotonicity”
    • Lines 400-407. Why are you now Spearman instead of Pareto? It should be if those weak correlations are due to random processes.

Author Response

> In the paper Quantifiers in Natural Language: Efficient Communication and Degrees of Semantic Universals the author analysis the language universals on quantifier words. For doing this they use a map languages on a set of experiments comparing complexity and comunicative cost. Their results suggest that both that efficient communication shapes semantic typology in both content and that semantic universals may not stand in need of independent

> first of all I would like to congratulate the author for this interesting article. Some parts of the article I have not reviewed in depth due to lack of sufficient knowledge of the specialty. However, I have some questions that I would like to ask.

Many thanks for the apt summary and kind words, and for your review more generally.  The paper has certainly improved in response thereto.

> Introduction

> Probably the introduction could be improved, considering more explanations and references. Particularly:

> “While the languages of the world vary greatly, linguists have discovered many restrictions on possible variation”. What restrictions? Could you put some examples? Are physiological constrains considered here (density of the air, size of lunges, etc)?

Thank you.  I have added some examples from phonology and from syntax, and then split the first paragraph into two.  The new second paragraph (about semantic universals in particular) also includes an example up-front. 

> Lines 20-26. Good explanation. I suggest to cite Elements of Information theory” from Joy A. Thomas y Thomas M. Cover. The concept of entropy could also be introduced.

Thanks; I have added that reference (and one more) here.  It seems too early to me to explicitly introduce entropy at this point, but I have made clear that it will be explicated in information-theoretic terms later in the paper.

> “The general claim: the semantic systems of the world’s languages optimally balance these two competing pressure” You could probably refer to this paper “Least effort and the origins of scaling in human language” Cancho and Sole; where it is shown that languages should lie close to a phase transition.

Thank you for drawing my attention to this wonderful paper.  I have added a footnote in the introduction mentioning it since the trade-off in that paper is highly relevant, though not exactly the same.

> Other universals could be introduced, for example recently. “Hahn, M., Jurafsky, D., & Futrell, R. (2020). Universals of word order reflect optimization of grammars for efficient communication. Proceedings of the National Academy of Sciences, 117(5), 2347-2353.” .

Thank you for pointing out this paper; I have also added a reference to it in the first paragraph.

> Linguistic laws are also universals (probably those that have been quantitatively measured in a more massive way), where probably the most popular is Zipf-law which has been found in all languages and predicted in “Zipf, G. K. (2016). Human behavior and the principle of least effort: An introduction to human ecology. Ravenio Books.” Those linguistic laws have been further found and studied in texts (Altmann, E. G., & Gerlach, M. (2016). Statistical laws in linguistics. In Creativity and universality in language (pp. 7-26). Springer, Cham.), speech (Torre, I. G., Luque, B., Lacasa, L., Kello, C. T., & Hernández-Fernández, A. (2019). On the physical origin of linguistic laws and lognormality in speech. Royal Society open science, 6(8), 191023.), in microscopic scales of human voice (Torre, I. G., Luque, B., Lacasa, L., Luque, J., & Hernández-Fernández, A. (2017). Emergence of linguistic laws in human voice. Scientific reports, 7(1), 1-10.) and even animal communication (Heesen, R., Hobaiter, C., Ferrer-i-Cancho, R., & Semple, S. (2019). Linguistic laws in chimpanzee gestural communication. Proceedings of the Royal Society B, 286(1896), 20182900.).

I have added a footnote in the introduction pointing to a more thorough discussion of linguistic laws in the general discussion section, which has been lengthened and re-organized accordingly.  

> Lines 151-160. References are needed. This paragraph could be explained in terms of Principle of least effort (Zipf) or similarly in terms of Information theory (Cover Thomas or Shannon).

I have added a footnote with these and other references, and pointed out that the _precise_ definitions in this paper will come in the next section.

> “Pareto frontier”. The search of Pareto optimal solutions should be better introduced.

I have added a sentence to the second paragraph of Section 2.4 elaborating the nature of /explaining the Pareto frontier.  The search for these solutions occurs is described in Section 3.2 and Appendix A.

> It would be interesting to plot optimality versus Degree of naturalness to check if there is indeed a linear correlation or maybe some other type of function between those variables. The fulfillment of Pearson correlation hypothesis should also be checked (lines 283-285).

I have included the requested plot and a brief discussion in an appendix (so as not to interrupt the flow of the paper).  I am unsure what the reviewer means by checking the "fulfillment of Pearson correlation hypothesis": I measured the correlation and bootstrapped confidence intervals, to verify that there is indeed a correlation.

> Degree of monotonicity: As far as this reviewer known, there is not some kind of degree of monotonicity and from the mathematical point of view it would make no sense at all. Maybe another term should be chosen or scientific references should be reported to backward the term “degree of monotonicity”

Indeed, this notion of a _degree_ of monotonicity is one of the contributions of this paper.  I provide references to some earlier papers that I'm a co-author of where we also use this definition (see refs 38, 39 on line 352 in the original submission).  While the reviewer states that "from the mathematical point of view it would make no sense at all", the concept is given a precise definition in terms of normalized mutual information below line 351.  I also provide a theorem showing that monotone quantifiers have degree 1, and that the degrees measured in this way track intuitions, whereby "between 3 and 5" (a conjunction of monotone quantifiers) has a fairly high degree, and "an even number of" has a very low degree.  To my mind, these results together show that this property does provide a graded measure of the notion of monotonicity.  If the reviewer still disagrees, I can consider alternative names.

> Lines 400-407. Why are you now Spearman instead of Pareto? It should be if those weak correlations are due to random processes.

This was a typo that has now been fixed (they are also Pearson correlations in this second experiment).  Thank you for catching this!

Reviewer 3 Report

This paper describes an efficient communication analysis of prominent semantic universals in the domain of quantifiers.

I found the paper to be clear and well-written, and I think it is essentially ready for publication.

The primary weakness I see is that the utility measure has (or seems to have) a fair degree of arbitrariness, with limited justification for the proposed measure. It makes sense that there needs to be a measure accounting for non-exact measures, but one could imagine other measures that would appear to do similar things. In general, mutual information seems to be a popular utility measure in the literature in which the author situates the proposed account, and one could imagine information-theory-based utility measures here: For instance, how about a listener's expected negative future surprisal upon learning that some object is or isn't in some set? At least some discussion acknowledging this degree of freedom would be welcome.

Author Response

> This paper describes an efficient communication analysis of prominent semantic universals in the domain of quantifiers.

> I found the paper to be clear and well-written, and I think it is essentially ready for publication.

Many thanks for the kind words!

> The primary weakness I see is that the utility measure has (or seems to have) a fair degree of arbitrariness, with limited justification for the proposed measure. It makes sense that there needs to be a measure accounting for non-exact measures, but one could imagine other measures that would appear to do similar things. In general, mutual information seems to be a popular utility measure in the literature in which the author situates the proposed account, and one could imagine information-theory-based utility measures here: For instance, how about a listener's expected negative future surprisal upon learning that some object is or isn't in some set? At least some discussion acknowledging this degree of freedom would be welcome.

Thank you for calling attention to this degree of freedom and for your helpful suggestion.  I've added two paragraphs to the end of Section 3.1: one discussing your measure of utility as well as another discussing alternative possible measures of informativeness using information theory, and noting that there is a degree of freedom here that could be pursued in future work.

Round 2

Reviewer 1 Report

Dear author,

The author has made all the changes proposed in my first revision. I congratulate you for having brought your line of research in semantics closer to the multidisciplinary reader of Entropy and, especially, for the new section referring to linguistic laws, very suggestive. I encourage you to explore open connections with quantitative linguistics and information theory in the near future.

Please consider just a few minor comments:

*********minor comments**********

1.- Consider to include in the main text (line 340) the footnote 18. It is important for the scope of the results and not a simple footnote.

2.- Line 533. Typo: rankof -> rank of

3.- Line 572. I would remove 'linearly' from the sentence, as there could be non-linear approximations to the problem.

4.- Line 612. Consider change at the end "semantic domains" -> "linguistic domains", since, as the author (and Debowski, 2020) has previously pointed out, there could be connections with other linguistic levels (syntax, pragmatics...). The end is thus much more suggestive, in my opinion.

5.- Appendix:  typo, insert line between 644-645.

6.- Line 649 and footnote 25. Please indicate which figure is.

7.- References. Please correct surname of main author in references 5, 9, 65:

Cancho R.F.I. -> Ferrer-i-Cancho, R.

****************

Congratulations again for your very interesting work,

Reviewer 1

Author Response

I thank the reviewer for their kind words and am delighted that they are satisfied with the revisions.  Once again, the paper has undoubtedly improved from responding to the original review, and I look forward to continuing to explore the connection with quantitative linguistics and information theory, as suggested.  Thanks as well for another close read: I have incorporated all of the minor comments into this newest version.

Reviewer 2 Report

The paper can be accepted in its present form.